# Palbociclib-Induced Cellular Senescence Is Modulated by the mTOR Complex 1 and Autophagy

**DOI:** 10.3390/ijms24119284

**Published:** 2023-05-26

**Authors:** Angel Cayo, Whitney Venturini, Danitza Rebolledo-Mira, Rodrigo Moore-Carrasco, Andrés A. Herrada, Estefanía Nova-Lamperti, Claudio Valenzuela, Nelson E. Brown

**Affiliations:** 1Center for Medical Research, School of Medicine, University of Talca, Talca 3460000, Chile; acayo@utalca.cl (A.C.); whitneyventurini@gmail.com (W.V.); danitza.rebolledo.ibi@gmail.com (D.R.-M.); 2Institute for Interdisciplinary Research, Academic Vice Rectory, University of Talca, Talca 3460000, Chile; 3Department of Clinical Biochemistry and Immunohematology, Faculty of Health Sciences, University of Talca, Talca 3460000, Chile; rmoore@utalca.cl; 4Lymphatic and Inflammation Research Laboratory, Facultad de Ciencias de la Salud, Instituto de Ciencias Biomédicas, Universidad Autónoma de Chile, Talca 3467987, Chile; andres.herrada@uautonoma.cl; 5Molecular and Translational Immunology Laboratory, Department of Clinical Biochemistry and Immunology, Pharmacy Faculty, Universidad de Concepción, Concepción 4070386, Chile; e.novalamperti@udec.cl

**Keywords:** senescence, cancer, mTORC1, autophagy, palbociclib, senescence-associated secretory phenotype

## Abstract

Despite not dividing, senescent cells acquire the ability to synthesize and secrete a plethora of bioactive molecules, a feature known as the senescence-associated secretory phenotype (SASP). In addition, senescent cells often upregulate autophagy, a catalytic process that improves cell viability in stress-challenged cells. Notably, this “senescence-related autophagy” can provide free amino acids for the activation of mTORC1 and the synthesis of SASP components. However, little is known about the functional status of mTORC1 in models of senescence induced by CDK4/6 inhibitors (e.g., Palbociclib), or the effects that the inhibition of mTORC1 or the combined inhibition of mTORC1 and autophagy have on senescence and the SASP. Herein, we examined the effects of mTORC1 inhibition, with or without concomitant autophagy inhibition, on Palbociclib-driven senescent AGS and MCF-7 cells. We also assessed the pro-tumorigenic effects of conditioned media from Palbociclib-driven senescent cells with the inhibition of mTORC1, or with the combined inhibition of mTORC1 and autophagy. We found that Palbociclib-driven senescent cells display a partially reduced activity of mTORC1 accompanied by increased levels of autophagy. Interestingly, further mTORC1 inhibition exacerbated the senescent phenotype, a phenomenon that was reversed upon autophagy inhibition. Finally, the SASP varied upon inhibiting mTORC1, or upon the combined inhibition of mTORC1 and autophagy, generating diverse responses in cell proliferation, invasion, and migration of non-senescent tumorigenic cells. Overall, variations in the SASP of Palbociclib-driven senescent cells with the concomitant inhibition of mTORC1 seem to depend on autophagy.

## 1. Introduction

Cellular senescence is a unique form of cell cycle arrest that can be triggered in response to a wide variety of stimuli, both physiological and pathological [1]. Among stressful stimuli that can induce cellular senescence, telomere dysfunction, oncogene activation, persistent genotoxic damage, and therapeutic stress are the most consistent in different models [2]. Importantly, while cellular senescence was considered an adaptive response aimed to prevent the proliferation of cells at risk of neoplastic transformation [3], we now know that senescent cells participate in physiological processes as diverse as embryonic development, wound healing, and tissue repair [4,5,6]. Of note, the stable and irreversible proliferative arrest seen in senescent cells is established and maintained by at least two signaling pathways centered on tumor suppressor proteins, the Tumor Protein P53 (p53)/Cyclin-Dependent Kinase Inhibitor 1A (p21^CIP1^) and Cyclin-Dependent Kinase Inhibitor 2A (p16^Ink4a^)/Retinoblastoma (pRB) pathways [7], which ensure that cells at risk of neoplastic transformation are not perpetuated in tissues [8]. Therefore, like apoptosis, cellular senescence acts as a barrier that cells must overcome to achieve immortalization and transformation, which would explain the high frequency of disruption of these pathways in human cancers [9,10].

Even though human cancers often harbor defects in the above-mentioned signaling pathways—and may thus become partially or completely resistant to the cytotoxic effects of chemotherapy or radiotherapy [11]—cancer cells of various lineages can still implement a senescence program in response to chemotherapeutic drugs, a form of stress-induced senescence referred to as therapy-induced senescence (TIS) [12]. To date, there are countless examples supporting the therapeutic role of TIS [13]. Several targeted molecules capable of inducing senescence in vitro and in vivo have been approved by the FDA (Food and Drug Administration) for the treatment of cancer [14]. Within the group of kinase inhibitors, cyclin-dependent kinase 4/6 (CDK4/6) inhibitors, including Palbociclib and Abemaciclib, have been successfully used in the treatment of advanced estrogen receptor-positive (ER+) and human epidermal growth factor receptor 2 (HER-2)-negative breast cancers [15,16,17]. Nevertheless, the effects of those senescent cells that remain in the tumor microenvironment or the residual tumor tissue following pro-senescence therapy are not always innocuous [3,18,19]. Senescent cells can synthesize and secrete a vast array of bioactive molecules, a feature known as the “Senescence-Associated Secretory Phenotype (SASP)”. Secreted molecules include growth factors, cytokines, chemokines, proteases, and components of the extra-cellular matrix [20]. While some of these factors can maintain, or even promote, senescence in a paracrine manner [21], other SASP factors with documented proinflammatory action [22] can promote a pro-tumorigenic microenvironment that can lead to tumor recurrence or acceleration of tumor progression [23,24,25]. For example, SASP components may contribute to the induction of EMT (Epithelial–Mesenchymal Transition), facilitating the acquisition of migratory and invasive features in cancer cells of epithelial origin [26,27,28]. Thus, at least in some settings, TIS can be associated with recurrence, metastasis, and, in general, a worse prognosis [29,30]. These observations also highlight the fact that, to be effective, TIS must be accompanied by the removal of senescent cells. More broadly, the persistence of senescent cells and their SASPs likely contribute to cancer recurrence in those individuals previously treated with pro-senescence anti-cancer drugs [31,32].

Since the beneficial or detrimental effects of senescent cells, particularly those arising in the context of TIS, are determined at least by the persistence of the senescent cells and the composition of their SASPs [33,34,35,36], pharmacological approaches capable of modifying the survival of senescent cells in the context of TIS, or pharmacological approaches that target specifically the SASP, would be expected to improve current pro-senescence therapies. This view implies the need to understand the signaling circuits and subcellular processes that promote the survival of senescent cells and their pro-inflammatory SASPs. The mTOR (Mechanistic Target of Rapamycin) complex 1 (mTORC1) and the autophagy process have become particularly prominent among these circuitries and processes. It should be noted that both the inhibition of the mTORC1 and the inhibition of autophagy have been the focus of study in cancer [37,38,39]. In this sense, using pro-senescence anti-cancer drugs, combined with mTOR inhibitors and/or autophagy inhibitors, may have preponderant effects on the SASP and, therefore, on the tissue environment.

mTORC1 is a serine/threonine kinase [40] functionally located downstream of various protein complexes that sense a wide variety of signals, including those derived from the activation of growth factor receptors and the availability of amino acids, glucose, oxygen, and ATP [41,42]. mTORC1 promotes cell growth and survival through the regulation of protein synthesis and other biosynthetic processes, while limiting autophagy-mediated catabolism [43,44]. Not surprisingly, cancer cells often show activation of this complex [45]. Notably, in some models, mTORC1 activity is also necessary for the implementation of cellular senescence [46]. For example, mTORC1 inhibition in senescent cells can lead to a reduction in the expression and secretion of inflammatory cytokines, selectively blocking the translation of membrane-bound Interleukin 1 Alpha (IL1α) by reducing the transcriptional activity of Nuclear Factor Kappa B Subunit 1 (NF-kappa B) [47]. This, in turn, leads to a reduction in the expression and secretion of Interleukin 6 (IL-6) and Interleukin 8 (IL-8) [48]. While mTORC1 plays a role in the implementation of different aspects of the senescent phenotype, the degree to which senescence depends on mTORC1 activity varies depending on the modality of senescence and/or the cell lineage involved. For example, mTOR inhibition can lead to a reversal of the senescent phenotype (as occurs in oncogene-induced senescence, OIS, and models) [49], to a reduction in the viability of senescent cells (as occurs in models of senescence induced by CDK4/6 inhibition), or to more subtle changes in SASP profiles [50].

Among the cellular processes that could explain the persistence of senescent cells in tissues, autophagy has gained particular importance in recent years. Autophagy can be induced by drugs as a process in which the cellular recycling system is activated to remove damaged or harmful components [51]. More generally, autophagy is an evolutionarily conserved catabolic process in which subcellular components undergo lysosome-mediated degradation [52]. Although several reports indicate that an upregulation of autophagy accompanies the induction of cellular senescence [53], the role of autophagy in senescence depends on the cellular context [54].

Studies of OIS (oncogene-induced senescence) models have suggested that autophagy is necessary to implement the senescent phenotype [55]. In these models, the inhibition of autophagy delays the implementation of OIS [56,57]. It was later shown that autophagy in these models contributes to the recycling of amino acids and other metabolites that can be subsequently used by a lysosome-bound mTORC1, known as TOR Autophagy Spatial Coupling Compartment (TASCC), for the synthesis of SASP components, such as IL-6 and IL-8 [58,59]. Therefore, a catabolic process (autophagy) is coupled to an anabolic process (mTORC1-dependent protein synthesis) to effectively coordinate the production of SASP components [53,60]. While attractive, this mechanism has not been extended to other models of cellular senescence, such as TIS, in the context of cancer treatment [61,62]. Furthermore, the model must be reconciled with the fact that autophagy induction is often accompanied by mTORC1 inhibition in non-senescent cells subjected to metabolic stress [58,63,64]. In summary, while mTORC1 is a negative regulator of autophagy in most non-senescent cells, at least in certain types of senescent cells, autophagy can feed the mTORC1 complex, which is necessary for the implementation of the SASP [65,66].

Here, we examine the functional impact of mTORC1 inhibition, or the combined inhibition of mTORC1 and autophagy, on cellular models of Palbociclib-induced senescence. We show that mTORC1 inhibition exacerbates the senescent phenotype induced by Palbociclib, a phenomenon that is autophagy-dependent. Furthermore, mTORC1 inhibition, and the co-inhibition of mTORC1 and autophagy, significantly affect the composition of factors produced and secreted by Palbociclib-driven senescent cells. These variations in SASP composition are also functionally relevant. For example, senescent cells with mTORC1 inhibition display SASPs that are more pro-invasive and promigratory than SASP profiles produced by senescent cells without mTORC1 inhibition. Surprisingly, the SASP from Palbociclib-driven senescent cells reduced the accumulation of cancer cells compared to the SASP from non-senescent control cells. This “antiproliferative effect” was exacerbated when the SASP from Palbociclib-driven senescent cells with mTORC1 and autophagy inhibition was tested. Overall, our results support a model in which senescence driven by CDK4/6 inhibition, combined with the inhibition of mTOR and autophagy, could lead to the appearance of senescent cells producing SASP profiles that are not only antiproliferative but also pro-migratory and pro-invasive, highlighting the caveats of modulating senescence phenotypes.

## 2. Results

### 2.1. Palbociclib-Induced Senescence in AGS and MCF-7 Cell Lines

As in vitro models of cellular senescence, AGS cells (derived from a human gastric adenocarcinoma) and MCF-7 cells (derived from a human mammary carcinoma) were exposed for 96 h to Palbociclib, a targeted drug that induces cellular senescence as a result of the inhibition of CDK4/6 kinases. These kinases, once activated by D-type cyclins, contribute to the phosphorylation-mediated inactivation of the retinoblastoma protein (pRB), an event necessary for the G1-to-S cell cycle transition [67,68]. Therefore, changes in the levels of pRB could affect the orchestration of cellular senescence in response to Palbociclib, as reported previously [69,70]. To confirm the expression of pRB in AGS and MCF-7 cells, the presence or absence of this protein was first assessed by indirect immunofluorescence (Appendix A). As shown in Appendix A, pRB was readily detectable in these cell lines. It should be noted that, according to the information reported by the cBioPortal for cancer genomics, the human *RB1* gene, as well as the genes encoding CDK4 (*CDK4*), CDK6 (*CDK6*), and cyclin D1 (*CCND1*), are all expressed in AGS and MCF-7 cells, without mutations reported so far (Appendix A).

Expectedly, exposure of both cell lines to Palbociclib led to decreased levels of phosphorylation at Serine-780 of pRB (phospho-Ser780-pRB), a catalytic reaction that depends on CDK4 or CDK6 (Figure 1A). Accordingly, the mRNA levels of the S-phase genes encoding the cell proliferation nuclear antigen (PCNA), dihydrofolate reductase (DHFR), and component 3 of the chromosome maintenance complex (MCM3) were all downregulated in AGS and MCF-7 cells upon a 96 h exposure to Palbociclib (Figure 1B). Importantly, the induction of cellular senescence was documented in both AGS and MCF-7 cells following the detection of in situ β-Galactosidase activity at pH 6.0 (Senescence-Associated β-Galactosidase Activity, SA-β-Gal; Figure 1C). Thus, there was an increase in the proportion of blue-stained SA-β-Gal positive cells in response to Palbociclib when compared to cells treated with the vehicle (Figure 1C).

It is worth mentioning that the doses of Palbociclib used for the induction of cellular senescence (1 μM for AGS cells and 0.5 μM for MCF-7 cells) were selected based on trypan blue-based viability assays (Appendix A), as well as the measurement of the levels of SA-β-Galactosidase activity determined by alternative, fluorescence-based, and more quantitative methods (Appendix A). Indeed, the doses selected were those capable of inducing senescence but, at the same time, preserving overall cell viability (Appendix A). Data collected on cell counts using trypan blue reflect the number of live cells adhered to the culture dish after incubation with Palbociclib for a specified time. As seen in Appendix A, increasing the concentrations of Palbociclib results in lower cell numbers compared to the control (0.02% DMSO-containing medium), which may be attributed to the increased numbers of senescent cells (Appendix A). To rule out the possibility that the selected doses of Palbociclib could be inducing cell death, apoptosis and/or necrosis levels were assessed by flow cytometry (Appendix A). Based on these experiments, treatment of AGS and MCF-7 cells for 96 h with 1.0 μM and 0.5 μM Palbociclib, respectively, did not lead to significant increases in the proportions of apoptotic or necrotic cells, compared to those cells (AGS and MCF-7) treated with the vehicle (Appendix A).

These experiments confirmed that Palbociclib effectively inhibits CDK4/6 activity in our cell models, leading to the induction of cell cycle arrest and cellular senescence, with overall preservation of cell viability.

### 2.2. Palbociclib-Induced Cellular Senescence Results in Reduced Levels of mTORC1 Activity

We next set out to assess the functional status of the mTORC1 complex in Palbociclib-driven senescent cells. As already mentioned, mTOR activity seems to play a major role in senescence induction and the implementation of senescent phenotypes [71,72,73]; although, this role can vary depending on the model of cellular senescence utilized. Importantly, we first confirmed the presence of the mTOR protein in AGS and MCF-7 cells using indirect immunofluorescence (Appendix A). Further, based on information reported in the cBioPortal for cancer genomics portal (Appendix A), the genes coding for mTOR protein and Raptor (mTORC1 complex activating protein) proteins do not present mutations in MCF-7 or AGS cells.

To examine the activity of mTORC1 in Palbociclib-driven senescent AGS and MCF-7 cells, phospho-residues of the target proteins of mTORC1, 4EBP-1, and p70S6 were detected by immunoblotting. Both proteins participate in protein synthesis, being phosphorylated by the mTORC1 at specific residues [74,75]. We found that the activity of the mTORC1 complex decreased in senescent AGS and MCF-7 cells (Figure 1D); although, the phosphorylation of p70S6 did not reach significance when densitometric analyses were carried out (Figure 1D). Based on these results, and in contrast to other models of cellular senescence [76,77], we concluded that Palbociclib-induced senescence in AGS and MCF-7 cells is not accompanied by an increase in the activity of mTORC1. Rather, a mild decrease in mTORC1 activity was evident in Palbociclib-driven senescent cells.

### 2.3. Further Inhibition of the mTORC1 Complex Exacerbates the Senescent Phenotype Induced by Palbociclib

To further study the role of mTORC1 in Palbociclib-induced senescence, we set out to inhibit the complex in Palbociclib-driven senescent AGS and MCF-7 cells using Rapamycin. The dose of Rapamycin chosen for these experiments (0.05 μM) corresponded to the minimal dose required to inhibit mTORC1 without eliciting a senescence response (Appendix A). While this dose of Rapamycin did slightly impair proliferation (Appendix A), this response was not due to an increase in the rates of cell death (Appendix A). Following these optimization experiments, we proceeded to expose AGS and MCF-7 cells to Palbociclib, or a combination of Palbociclib and Rapamycin, for 96 h. SA-β-Gal assays revealed that the combined treatment significantly increased the number of senescent cells, albeit slightly, compared to treatment with Palbociclib alone (Figure 2A). Accordingly, the number of cells treated with the combination of drugs that remained adhered to the culture dish towards the end of the incubation period was also reduced, compared to treatment with Palbociclib alone (Figure 2B), suggesting an exacerbation of the senescence response in senescent cells with the inhibition of mTORC1. Interestingly, this increase in senescent cells after treatment with Palbociclib and Rapamycin was accompanied by a further reduction in the phosphorylated levels of pRB at Serine-780, an effect that was particularly marked in MCF-7 cells (Figure 2C). Nevertheless, this trend did not reach significance when compared with the levels of phospho-pRB in cells treated with Palbociclib alone (Figure 2C). Therefore, mTORC1 inhibition seems to exacerbate the pro-senescence effect of Palbociclib.

### 2.4. Blockade of Autophagy in Palbociclib-Driven Senescent Cells with Inhibition of mTORC1 Reverses the Senescent Phenotype

It has been well established that the inhibition of the mTORC1 complex, either drug-induced or secondary to starvation, leads to increased rates of autophagy, a cellular process that maintains cell viability under metabolically stressful conditions [63,78,79]. Interestingly, autophagy is also upregulated in at least some models of cellular senescence, where it is believed to help implement senescent phenotypes, including the SASP [56,80]. So far, however, the interplay between this senescence-associated autophagy and the functional status of the mTORC1 complex is far from clear. Based on our results (Figure 1 and Figure 2), Palbociclib-induced senescence, and the exacerbation of the senescent phenotype in cells simultaneously treated with Palbociclib and Rapamycin, is expected to occur with increasingly high levels of autophagy. To determine the effect of autophagy inhibition in Palbociclib-driven senescent cells in which mTORC1 had been also inhibited, we took advantage of the drug Spautin-1 (6-Fluoro-N-(4-fluorobenzyl) quinozaline-4-amine), a compound that inhibits the activities of ubiquitin-specific peptidases, key in the autophagy process [81]. Importantly, we carried out several experiments aimed to determine the minimal dose of Spautin-1 necessary to produce a sustained inhibition of autophagy (Appendix A), without compromising cell viability (Appendix A) through the induction of cell death (Appendix A), and without inducing per se cellular senescence (Appendix A). Based on these and other results, we selected 1 μM as the concentration of Spautin-1 for the next experiments.

Interestingly, the reduced levels of phosphorylation at Serine 780 of pRB observed in Palbociclib-driven senescent cells (Figure 1A) were further reduced in senescent cells treated with Rapamycin, or with Rapamycin and Spautin-1 (Figure 3A). As expected, Palbociclib-driven senescent cells exposed to Rapamycin showed even lower levels of mTORC1 activity when compared to senescent cells driven by Palbociclib only (Figure 2D and Figure 3B). Interestingly, senescent cells treated with Rapamycin and Spautin-1 failed to reverse this hypo-phosphorylation effect (Figure 3B). 

One of the most common assays to assess autophagy is the detection of LC3 levels [82]. The microtubule-associated protein MAP1LC3 (LC3) has two isoforms: LC3-I, primarily cytosolic, and LC3-II, which is conjugated to phosphatidylethanolamine (PE) and is present in autophagosomes [83]. Consequently, as a general pattern, the level of LC3-I decreases and the level of LC3-II increases upon autophagy induction [84]. On the other hand, SQSTM1/p62 is one of the best-known autophagy substrates. This protein acts as a bridge between LC3 and polyubiquitinated cargoes destined for degradation. Since p62 is mainly degraded during autophagy [85], the joined measurement of p62 and LC3 provides a better view of autophagic flux.

We next assessed the levels of autophagy in our cellular models of Palbociclib-induced senescence. As shown in Figure 3C, an upregulation in autophagic flux in Palbociclib-driven senescent cells was evidenced by a decrease in LC3-I and an increase in LC3-II levels. As expected, the inhibition of the mTOR in senescent cells led to a further increase in the rates of autophagy, a trend that was reversed by the addition of Spautin-1 (Figure 3C). Of note, SQSTM-1/p62 levels confirmed our findings. Thus, p62 levels were reduced in Palbociclib-driven senescent cells, and were even lower in senescent cells with the inhibition of mTORC1. Nonetheless, p62 levels did not achieve equilibrium when autophagy was inhibited (Figure 3D).

Furthermore, the levels of phosphorylation at Ser-757 of ULK1, indicative of autophagy inhibition, were reduced in Palbociclib-driven senescent cells (Figure 3E). Expectedly, the inhibition of mTORC1 in these cells further decreased ULK1 phosphorylation. Importantly, ULK1 acts as a point of convergence for multiple signals that control autophagy [86,87]. In particular, mTORC1 phosphorylates ULK1 at Ser-757, disrupting the interaction between ULK1 and AMPK, and thus blocking autophagy [88].

We next tested the cellular effects of inhibiting mTORC1 and autophagy in Palbociclib-driven senescent cells. As shown in Figure 4A, while the proportion of AGS and MCF-7 cells in G1-G0 was not different between Palbociclib-treated senescent cells and Palbociclib-treated senescent cells with mTORC1 and autophagy inhibition, the number of adherent cells was progressively reduced with the combined treatments. Indeed, the decrease in the accumulation of cells was most significant in Palbociclib-driven senescent cells treated with mTORC1 and Spautin-1 (Figure 4B). Interestingly, the proportion of cells positive for the SA-β-Gal staining was reduced in Palbociclib-treated cells with mTORC1 and autophagy inhibition in comparison to Palbociclib-treated cells with mTORC1 inhibition only, returning to levels of senescence observed in cells treated with Palbociclib only (Figure 4C). A similar tendency was observed when SA-β-Galactosidase activity was detected by fluorometry (Figure 4D).

Based on these results, we can conclude that the inhibition of the autophagic flux in senescent cells with reduced mTORC1 activity reverses the senescent phenotype to levels observed in cells treated with Palbociclib only, indicating that the exacerbation of senescence observed in cells treated with Palbociclib and Rapamycin is dependent on autophagy induction.

### 2.5. The Levels of Selected Factors Secreted by Senescent Cells Change upon Inhibition of mTORC1 and/or Autophagy

To examine differences in the levels of selected factors secreted by senescent cells deficient in mTORC1 activity, or deficient in mTORC1 activity and autophagy, conditioned media were collected from senescent cells subjected to mTORC1 and/or autophagy inhibition. Interestingly, the amount of total protein secreted by senescent AGS and MCF-7 cells was reduced compared to the amount of total protein secreted by non-senescent (control) cells (Figure 5A). This decrease was accentuated in the case of senescent cells with mTORC1 inhibition but returned to “senescent” levels when autophagy was blocked (Figure 5A).

Next, we set out to quantify selected factors secreted by Palbociclib-driven senescent cells and by Palbociclib-driven senescent cells subjected to either the inhibition of mTORC1 or combined inhibition of mTORC1 and autophagy. To this end, we used a flow cytometry-based bead array to detect selected cytokines. As explained in the Section 4, the concentrated conditioned media (30-fold) were evaluated one time by the bead-based assay. Nevertheless, these experiments revealed differences in the secretion patterns of senescent AGS and MCF-7 cells, as well as differences in the way in which mTORC1 inhibition, or the combined inhibition of mTORC1 and autophagy, impinged on the secretome (Figure 5B). Surprisingly, while senescent AGS cells were capable of secreting IL-12p70, IL-10, and IL-8, their levels were lower than those secreted by non-senescent (control) AGS cells. Notably, these cytokines dropped even further in conditioned media derived from senescent AGS cells deficient in mTORC1 activity. That this further reduction was dependent on autophagy induction was evident from the fact that autophagy blockade restored the levels of these cytokines to those observed in conditioned media from Palbociclib-driven senescent AGS cells (Figure 5B). In contrast, TNF-a, IL-6, and IL-1b were almost exclusively detectable in conditioned media from Palbociclib-driven senescent AGS cells but not in conditioned media derived from senescent cells mTORC1, or mTORC1 and autophagy, which were blocked.

In contrast, Palbociclib-driven senescent MCF-7 cells secreted higher amounts of IL-12p70, IL-8, IL-6, TNF-a, and IL-1b compared to their non-senescent (control) counterparts (Figure 5B). Similar to senescent AGS cells, the levels of cytokines secreted by Palbociclib-driven senescent MCF-7 cells were modified by the inhibition of mTORC1 and autophagy (Figure 5B). Thus, except for IL-10, which was not detectable in any of the conditioned media produced by senescent MCF-7 cells, cytokines were detected in similar or higher amounts in conditioned media derived from Palbociclib-driven senescent MCF-7 cells when compared with conditioned media derived from non-senescent (control) MCF-7 cells (Figure 5B). Interestingly, the amounts of cytokines were even higher in the media derived from Palbociclib-driven senescent MCF-7 cells with the inhibition of mTORC1; autophagy blockade in these cells restored the levels of cytokines to those observed in senescent MCF-7 cells (Figure 5B).

Together, these experiments confirmed the known variabilities in cytokine secretion patterns displayed by senescent cells of different cell lineages. Indeed, we found that some cytokines were reduced in media derived from senescent AGS cells but increased in media derived from senescent MCF-7 cells. Despite this, the inhibition of mTORC1 tended to exacerbate these responses, further reducing secretion in senescent AGS cells or further increasing secretion in senescent MCF-7 cells. Interestingly, these responses were reversed by Spautin-1, demonstrating that they were dependent on autophagy activation in the context of mTORC1 inhibition, and highlighting the mTORC1–autophagy axis as an essential regulator of SASP components in different cell lineages.

### 2.6. The SASP Derived from Palbociclib-Driven Senescent Cells with mTORC1 Inhibition Enhances the Pro-Invasive and Pro-Migratory Capabilities of Cancer Cells

After documenting the changes in the secretion profiles of senescent cells with different statuses of mTORC1 and autophagy activity, we next assessed the pro-migratory and pro-invasive effects that factors secreted by these senescent cells have on their non-senescent counterparts. First, non-senescent (actively proliferating) cells were exposed to conditioned media derived from Palbociclib-driven senescent cells, with or without mTORC1 inhibition or the combined inhibition of mTORC1 and autophagy. As shown in Figure 6A, the total number of non-senescent AGS or MCF-7 cells was reduced following their exposure for 72 h to conditioned media derived from different senescent cells. This antiproliferative effect was particularly marked in cells that had been exposed to a conditioned medium derived from Palbociclib-driven senescent cells with the simultaneous inhibition of mTORC1 and autophagy (Figure 6A; compare the effect of media obtained from cells previously treated with Palbociclib, Rapamycin, and Spautin-1 with the effect of conditioned media obtained from cells treated with the vehicle, DMSO). These data suggest that the SASP of senescent cells in which mTORC1 and autophagy had been blocked acquires antiproliferative properties. It is worth mentioning that the conditioned media (secretomes) of senescent AGS and MCF-7 cells with mTORC1 inhibition, or with combined mTORC1 and autophagy inhibition, were collected 24 h after the end of drug treatment (96 h). After drug exposure, AGS and MCF-7 cells were washed 3 times with 1×PBS, and the necessary amount of serum-free medium was added to each culture dish. Therefore, the secretomes analyzed are free of any residual drugs or FBS, factors that might interfere with the outcome of these experiments.

Next, we assessed the pro-invasive capabilities of non-senescent cells after being exposed to different conditioned media. To this end, Transwell chambers were used. As shown in Figure 6B, the pro-invasive capabilities of non-senescent AGS and MCF-7 cells were enhanced by conditioned media derived from the respective Palbociclib-driven senescent cells. Notably, this pro-invasive effect was even more significant when conditioned media derived from senescent cells with reduced mTORC1 activity was tested (Figure 6B). Interestingly, invasion enhancement was reversed when non-senescent cells were exposed to conditioned media derived from Palbociclib-driven senescent cells with the simultaneous inhibition of mTORC1 and autophagy.

Finally, the migratory capacity of non-senescent cells following their exposure to conditioned media derived from Palbociclib-driven senescent cells was assessed by wound healing assays. Due to differences in the dynamics of wound closure, the endpoint times chosen were 24 h and 48 h for AGS and MCF-7 cells, respectively. As shown in Figure 6C,D, consistent with the invasion assays, non-senescent cells tended to close the wound sooner when exposed to conditioned media derived from senescent cells. This effect was even more evident when the media derived from senescent cells with reduced mTORC1 activity was tested (Figure 6C). Interestingly, this enhancing effect was reversed when the media used was derived from senescent cells with the simultaneous inhibition of mTORC1 and autophagy.

In summary, the functional features of SASPs produced by senescent AGS and MCF-7 cells appear to be regulated, at least in part, by the activity of mTORC1 and autophagic flux. Changes in the SASP secondary to the inhibition of mTORC1 and/or autophagy led to diverse proliferative, migratory, and invasive outputs when tested on non-senescent cells. In particular, the pro-invasive and pro-migratory effects of SASPs derived from senescent cells with mTORC1 inhibition seemed autophagy-dependent.

## 3. Discussion

In this work, we studied the functional consequences of inhibiting mTORC1, or mTORC1 and autophagy, on senescent cells driven by Palbociclib, an inhibitor of CDK4/6. To this end, we took advantage of MCF-7 (breast carcinoma) and AGS (gastric adenocarcinoma) cells, both of which retain a functional pRB pathway and are responsive to CDK4/6 inhibition. While the initial development or assessment of anti-cancer drugs requires the use of cell lines, animal models, especially patient-derived xenograft mouse models, are an essential bridge between preclinical studies and clinical trials in cancer treatment [89]. In this context, we are aware of the limitations of using an in vitro model and we recognize the need to confirm our findings using in vivo models. Nevertheless, our results clearly suggest the therapeutic relevance of modulating drug-induced senescence.

Cellular senescence has traditionally been considered a form of stress response that, in the context of cancer, may serve as a mechanism through which anticancer drugs can halt tumor growth [90]. Pro-senescence anticancer therapies of clinical relevance can be exemplified by the recently developed cyclin-dependent kinases 4 and 6 (CDK4/6) inhibitors that interfere with the phosphorylation of the retinoblastoma protein (pRB) during the G1-S transition [91]. Palbociclib (PD 0332991), the prototypic CDK4/6 inhibitor, results in the arrest of cell-cycle progression and senescence induction in various in vitro and in vivo models [92,93,94,95]. Presently, Palbociclib is used in the treatment of patients with disseminated breast cancers that are positive for estrogen receptors [96], as well as patients with advanced gastric or esophageal cancer [97]. Despite their therapeutic benefits, however, senescence cells generated in the context of Palbociclib-induced senescence may be capable of remodeling the tissue microenvironment through the secretion of numerous bioactive molecules, including cytokines and chemokines with demonstrated pro-inflammatory activity [20,35], a feature known as the senescence-associated secretory phenotype (SASP) [23,98,99]. Thus, the post-therapy persistence of senescence cells, with their SASP-mediated paracrine actions, could lead, paradoxically, to cancer recurrence or accelerated tumor growth [24,100,101,102]. More broadly, these caveats have underscored the need to better characterize signaling pathways and processes involved in the orchestration of therapy-induced senescence [23,61,77] (TIS). One can envision that, by pharmacologically modulating senescent phenotypes, including the SASP, it would be possible to maximize and minimize the anti-cancer and detrimental effects, respectively, of persistent senescent cells. Among signaling pathways and processes that have been described in senescent cells, and whose modulation could alter senescent phenotypes, the mTOR-centered pathway and the process of autophagy are particularly amenable to pharmacologic modulation [73,103,104,105,106]. In the present study, we explored the consequences of blocking the mTORC1 complex, autophagy, or both, in Palbociclib-driven senescent cells. To this end, the human gastric adenocarcinoma cell line AGS and human mammary adenocarcinoma cell line MCF-7 were used. Both cell lines have been used by various research groups as in vitro models of cellular senescence [80,92,107,108]. Senescence induction was carried out here by exposing AGS and MCF-7 cells to Palbociclib for 96 h.

Studies carried out in several models of cellular senescence initially indicated that the mTORC1 complex was mostly active in senescent cells [46]. Indeed, the implementation of cellular senescence was delayed by the inhibition of mTORC1 in models of oncogene-induced [49,57] and replicative senescence [109]. Contrary to these reports, however, we found that Palbociclib-driven senescent AGS and MCF-7 cells displayed a partial but significant decrease in mTORC1 activity, evidenced by reduced levels of phospho-4EBP-1 but not phospho-p70S6K. This discrepancy may reflect differences in the cellular system used (primary versus fully transformed cells), the senescence-inducing stimulus (oncogene activation versus the inhibition of CDK4/6), or both. Alternatively, based on mTOR’s role in integrating upstream signals that detect the availability of metabolic substrates and growth-promoting signals, it is tempting to speculate that CDK4/6 inhibition might be somehow mimicking the lack of extracellular growth factors, leading to reduced mTORC1 activity. Additionally, it would not be unreasonable to evaluate the activity levels of mTOR complex 2 (mTORC2) in this model since it is well known that mTORC2 participates in the reorganization of the cellular cytoskeleton and, therefore, could modulate the morphology of senescent cells.

Interestingly, the further Rapamycin-mediated inhibition of the mTORC1 activity in Palbociclib-driven senescent MCF-7 and AGS cells resulted in a higher proportion of senescent cells, reflected in an increased number of senescent cells and a greater intensity of the SA-β-Gal activity per cell, as well as a reduced accumulation of cells over time. At least in MCF-7 cells, the exacerbation of the senescent phenotype was accompanied by a greater decrease in the phosphorylation of pRB at Serine-780, suggesting that mTORC1 inhibition could enhance the inhibitory effect of Palbociclib on CDK4/6, at least in some cell types. In this context, mTORC1 activation, particularly the activation that is dependent on growth factor availability, is known to lead to cyclin D1 synthesis [110]. Therefore, it is possible that our data reflect the combined effect of CDK4/6 inhibition and reduced D-type cyclin synthesis. Future studies will be necessary to confirm this hypothesis.

In addition to increased levels of mTORC1 activity, some senescent cells often display high rates of autophagy [73,80]. Accordingly, we found that Palbociclib-driven AGS and MCF-7 cells display increased levels of autophagy as well (Figure 3). Since the activation of mTORC1 negatively modulates autophagy, its induction in senescent cells must be independent of mTOR’s functional status, at least in some models of senescence. Despite this, a causal link between autophagy and senescence has been proposed [53,54,111]. Thus, the blockade of autophagy in human fibroblasts can delay the onset of oncogene-induced senescence or abrogate senescent phenotypes [57]. In this setting, autophagy would generate a high flux of recycled amino acids that can activate lysosome-bound mTORC1, a step necessary for the synthesis of SASP factors such as IL-6 and IL-8 [59,112]. Therefore, while mTORC1 is an upstream regulator of autophagy in most cells, at least in certain types of senescent cells, autophagy can feed the mTORC1 complex, a step necessary to implement the SASP [65,66]. Accordingly, mTORC1 inhibition may suppress the expression and secretion of inflammatory cytokines in senescent cells by selectively blocking the translation of membrane-bound IL-1α and by reducing the transcriptional activity of NF-kappa β [47], which, in turn, leads to a reduction in the expression and secretion of IL-6 and IL-8 [48].

Expectedly, the upregulation of autophagy in Palbociclib-driven senescent AGS and MCF-7 cells was exacerbated by the further inhibition of mTORC1. Notably, Palbociclib-driven senescent AGS and MCF-7 cells that were additionally subjected to combined mTORC1 and autophagy inhibition showed levels of cellular senescence similar to those found in Palbociclib-driven senescent cells, suggesting that the exacerbation of senescence observed in senescent cells with mTORC1 inhibition was dependent on autophagy. These changes in the levels of senescence upon mTORC1 and/or autophagy inhibition were mirrored by changes in the levels of selected components of the SASP when conditioned media were analyzed using a flow cytometry-based bead array. For example, the upregulation in the levels of IL-12p70, IL-8, IL-6, TNF-α, and IL-1β found in conditioned media derived from Palbociclib-driven senescent MCF-7 cells was even higher in conditioned media derived from Palbociclib-driven senescent MCF-7 cells with mTORC1 inhibition. However, autophagy blockade in these cells restored the levels of cytokines to those observed in Palbociclib-driven senescent MCF-7 cells. Taken together, these experiments confirm the known variabilities in the patterns of cytokine secretion displayed by senescent cells of different cell lineages and highlight the mTORC1–autophagy axis as an essential regulator of SASP components across different senescent cells. Nevertheless, the variations in the SASP components must be confirmed by more precise quantitative methods such as proteomics and the variations in gene expression profiles by transcriptional analysis.

Finally, we explored the paracrine effects of senescent cells on their non-senescent counterparts. Interestingly, conditioned media derived from senescent cells (Palbociclib-treated), senescent cells with mTORC1 inhibition (Palbociclib- and Rapamycin-treated), or senescent cells with inhibition of mTORC1 and autophagy (Palbociclib-, Rapamycin- and Spautin-1-treated) produced a progressive decrease in the proliferative capacity of non-senescent cells. Nevertheless, non-senescent cells of either lineage became more migratory and invasive upon exposure to factors released by Palbociclib-driven senescent cells, which was even more evident when media derived from senescent cells with mTORC1 inhibition were utilized. Notably, this enhancement of pro-migratory and pro-invasive capabilities was, again, abrogated when conditioned media derived from senescent cells with the concomitant inhibition of mTORC1 and autophagy were used in these assays. Thus, it seems that the inhibition of mTORC1 in Palbociclib-driven senescent cells results in SASP profiles with enhanced pro-migratory and pro-invasive capabilities, a phenomenon that seems to be autophagy-dependent. Our results also show that the proliferative effects of senescent cells can be dissociated from their pro-migratory/invasive effects.

Overall, our findings demonstrate the relevance of redirecting the secretory profiles of senescent cells through pharmacological intervention. More specifically, the fact that some functional features of Palbociclib-driven senescent cells, including the SASP, can be modulated by drugs that block mTORC1 or autophagy, offers new ways to enhance the beneficial, anti-proliferative, effects of inhibiting CDK4/6 in tumors with molecular evidence of high CDK4/6 activity. At the same time, a note of caution must be added before these combinations of drugs can be introduced into clinical practice, as the secretome of Palbociclib-driven senescent cells could enhance the migratory and invasive capabilities of cancer cells that might have escaped senescence.

## 4. Materials and Methods

### 4.1. Cell Culture

Propagation of cells in culture was carried out according to the bioethical and biosafety regulations established by the University of Talca. All the experiments described in this work were performed on MCF-7 (mammary carcinoma; HTB-22, ATCC, Gaithersburg, MD, USA) and AGS (gastric adenocarcinoma; CRL-1739, ATCC) cells. Each cell line was propagated in sterile culture plates of 10 cm or 6 cm in diameter, or in 6, 12, or 24 multi-well plates, depending on the specific experiment. MCF-7 cells were cultured in high glucose DMEM (Dulbecco’s Modified Eagle Medium) medium (Hyclone, Thermo Scientific, South Logan, UT, USA), supplemented with human Insulin (10 µg/mL; Gibco, Thermo Scientific, South Logan, UT, USA). AGS cells were cultured in RPMI medium (Hyclone, Thermo Scientific, South Logan, UT, USA). Culture media for both cell lines were all supplemented with 10% Fetal Bovine Serum (FBS; Hyclone, Thermo Scientific, South Logan, UT, USA), 25 µg/mL Amphotericin B (InvivoGen, San Diego, CA, USA), 25 µg/mL Gentamicin (Hyclone, Thermo Scientific, South Logan, UT, USA), and 5 µg/mL Plasmocin (InvivoGen, San Diego, CA, USA).

### 4.2. Induction of Cellular Senescence

To induce cellular senescence, cells were exposed to the inhibitor of cyclin-dependent kinases 4 and 6 (CDK4/6) Palbociclib (PD0332991, Pfizer, New York, NY, USA). To this end, MCF-7 cells were exposed to 0.5 μM Palbociclib, and AGS cells were exposed to 1.0 μM Palbociclib, for 96 h. In each case, the drug-containing medium was replaced after 48 h, with the objective of maintaining the activity of the drug. Doses and times of exposure to the drug were previously determined in optimization experiments. The drug vehicle DMSO (Dimethyl sulfoxide) was used for all the controls at a final concentration of 0.02%. Induction of senescence was confirmed through the histochemical detection of the activity of the β-Galactosidase enzyme at suboptimal pH (pH 6.0), essentially as previously described [56,80]. In addition, to corroborate the inhibition of CDK4/6 activity, the levels of phosphorylation of the protein pRB (retinoblastoma) at Serine 780 were determined (Cell Signaling # 9307, Danvers, MA, USA) by Western blotting.

### 4.3. Senescence-Associated β-Galactosidase Assay

Cells were seeded in triplicate at a density of 2 × 10^4^ cells per well in 12-well plates. To facilitate microscopic inspection of cells that were positive for the assay, glass coverslips (Corning, Merck, Darmstadt, Germany) were added to the bottom of the wells prior to cell seeding. Adherent cells were then exposed to Palbociclib, alone or in combination with other drugs, and washed twice with 1×PBS (Phosphate Buffered Saline) (Hyclone, Thermo Scientific, South Logan, UT, USA). After washing, cells were fixed with 2% Paraformaldehyde/0.2% Glutaraldehyde in 1×PBS for 10 min at room temperature. Fixed cells were washed twice for 5 min with 1×PBS at room temperature. Cells were then incubated in staining solution (40 mM Citric Acid/Sodium Phosphate pH 6.0; 5 mM Potassium Hexacyanoferrate (II) trihydrate; 5 mM Potassium Hexacyanoferrate (III); 150 mM Sodium Chloride; 2 mM Magnesium Chloride; 50 mg/mL X-Gal (5-bromo-4-chloro-3-indolyl-β-D-Galactopyranoside)) overnight in a moist chamber, at 37 °C, and in the dark. The next day the cells were washed briefly in a solution of 0.2% DMSO, followed by two washes with 1×PBS, 5 min each. Finally, the coverslips were mounted on slides using glycerol. Brightfield images were captured using a 40× objective (BX53 microscope, Olympus). For the analyses of positive cells, the Image-J Fiji program was used. Five independent fields were considered in which the total cells per field were quantified as 100% and the positive cells as a percentage of this.

### 4.4. Fluorometry- and Flow Cytometry-Based Assays for the Detection of Senescent Cells

When necessary, the amount of β-Galactosidase protein in senescent cells was also determined by fluorometric and cytometric approaches. For this purpose, cells were seeded in 6-well plates at a density of 5 × 10^4^ cells per well. After exposing the cells to different pharmacological treatments (induction of senescence with Palbociclib, inhibition of mTORC1 with Rapamycin, inhibition of autophagy with Spautin-1), the fluorescence produced by a commercial substrate of β-Galactosidase was detected either by flow cytometry (ENZ-KIT129-0120, Enzo Life Sciences, New York, NY, USA) or by in situ fluorometry in living cells (ENZ-KIT130, Enzo Life Sciences, New York, NY, USA), according to the manufacturer’s instructions.

### 4.5. Flow Cytometry Analyses

For cytometric analyses, cells were first exposed to each drug, or combination of drugs, for 96 h. Cells were trypsinized, physically removed from the plates, re-suspended in 1×PBS, and then fixed for 30 min at 4 °C in 70% Ethanol. After two washes with 1×PBS, cells were centrifuged at 850× *g* for 5 min and incubated with 100 μg/mL of Ribonuclease A. For cytometry-based cell cycle analyses, cells were exposed to 50 μg/mL of Propidium Iodide, while for the analyses of cell death, a Kit based on Alexa Fluor^TM^ 488-coupled Annexin V and Propidium Iodide was used, following the manufacturer’s instructions (Thermo Fisher Scientific # V132459). As stated above, flow cytometry was also used to quantify Senescence-Associated-β-Galactosidase activity in living cells (ENZ-KIT130-0010), according to the manufacturer’s instructions. All analyses were performed on a BD FACSCaliburTM flow cytometer (Becton Dickinson, Franklin Lakes, NJ, USA) under the supervision of Dr. Andrés Herrada (Universidad Autónoma de Chile).

### 4.6. Cell Counting

For all pharmacological experiments, cells were cultured in triplicate in 6-well plates at a density of 5 × 10^4^ cells per well, or in 100 cm diameter plates at a density of 5 × 10^5^ cells per plate. Cells were exposed to complete culture medium supplemented with different doses of a drug (or combination of drugs) for 96 h. In each case, the drug-containing medium was replaced after 48 h, with the objective of maintaining the activity of the drug. After washing with 1×PBS, cells were detached from the surface of the well or plate using 250 μL of Trypsin, and were then resuspended in 750 μL of fresh complete medium. The quantification of the total number of cells, as well as those cells that were positive for the exclusion test with Trypan Blue (4% solution), was performed with the Luna-II^TM^ automated counter (Logos Biosystems, Villeneuve d’Ascq, France).

### 4.7. Pharmacological Inhibition of the mTOR Complex 1 (mTORC1)

For the inhibition of mTORC1, senescent MCF-7 and AGS cells were exposed for 96 h to culture media supplemented with 50 nM Rapamycin (Rapamycin CAS 53123-88-9, Santa Cruz, Dallas, TX, USA), replacing the medium with the same dose of drug after 48 h in order to ensure maximum drug activity throughout the exposure time. The drug vehicle DMSO (Dimethyl sulfoxide) was used for all the control groups at a final concentration of 0.02%. mTORC1 inhibition was verified by detecting, through Western blotting, the levels of the following phosphorylated versions of molecular targets of mTORC1: Phospho-threonine 37/46 of 4E-BP1 (Cell Signaling # 2855), Phospho-serine 389 of P70S6K (Cell Signaling #9205), Phospho-serine 371 from P70S6K (Cell Signaling #9208).

### 4.8. Autophagy Inhibition

The drug Spautin-1 (Cas 1262888-28-7, Merck, Rahway, NJ, USA) was used to inhibit autophagy in senescent MCF-7 and AGS cells. To this end, senescent or non-senescent MCF-7 or AGS cells were exposed to 1 μM of Spautin-1 for 96 h, the medium being replaced with the same dose of drug at 48 h to ensure the maximum activity of the drug during the entire exposure time. The drug vehicle DMSO (Dimethyl sulfoxide) was used for all the control groups at a final concentration of 0.02%. Inhibition of autophagy was corroborated by Western blotting for the detection of proteins involved in the autophagy process, or whose degradation is mediated by autophagy, namely, LC3B (Cell Signaling # 3868) and SQSTM1/p62 (Cell Signaling #8025). Additionally, quantitative PCR was used to evaluate the expression of genes that code for proteins involved in the autophagy process (*Beclin-1* and *ULK-1*).

### 4.9. Protein Preparation and Quantification

For protein extraction, MCF-7 or AGS cells were cultured in 10 or 6 cm diameter dishes. Cells were collected by scratching in 1 mL of 1×PBS, and centrifuged at 5000 rpm for 5 min at 4 °C. The cell pellet was then re-suspended in 100 μL of RIPA buffer (150 mM Sodium Chloride; 1% Nonidet P-40; 0.5% DOC; 0.1% SDS; 50 mM Tris pH 7.4) supplemented with protease and phosphatase inhibitors (Merck # P8340 and # P5726, Darmstadt, Germany), and the mixture let stand for 30 min on ice. Finally, cell lysates were centrifuged at 15,000 rpm for 15 min and at 4 °C. The supernatant was collected in Eppendorf tubes and stored at −20 °C. The concentration of most protein samples was estimated using the Bradford assay (Merk, Darmstadt, Germany), using a known calibration curve for Bovine Serum Albumin (BSA) (0.5, 1, 2, 4, 8, and 10 mg). Alternatively, the Pierce^TM^ BCA Protein Assay Kit was used, according to the manufacturer’s instructions (Thermo Fisher Scientific #23225). In all cases, the total protein concentration was estimated from absorbance measured at 595 nm wavelength. The final concentration was determined after performing an interpolation of the values of the curve.

### 4.10. Protein Electrophoresis in Polyacrylamide Gels, SDS-PAGE

A volume corresponding to 10–50 μg of protein was combined with the appropriate volume of loading buffer (3×SDS sample buffer: 188 mM Tris-Cl pH 6.8; 3% SDS; 30% Glycerol; 0.01% Bromophenol Blue; 15% β-Mercaptoethanol), and the mixture was heated for 5 min at 95 °C. Proteins were separated on polyacrylamide gels (7.5–12%) at a fixed resistance of 25 milli-Amperes, and then electro-transferred to nitrocellulose membranes (UltraCruz sc-3724, Santa Cruz Biotechnology, Santa Cruz, CA, USA) in transfer buffer (25 mM Tris-HCl pH 7.6; 192 mM glycine; 20% methanol; 0.03% SDS) for 90 min at 100 Volts. Once the transfer was complete, the membranes were blocked for one hour at room temperature in a solution of 1% Tween-20 (*v*/*v*)/5% non-fat milk (*w*/*v*)/1×TBS. After blocking, membranes were incubated overnight at 4 °C, under constant agitation, with the corresponding primary antibody diluted in blocking solution. Next day, the membranes were washed 4 times, 7 min each, in 1% Tween-20 (*v*/*v*)/1×TBS, under constant agitation. The membranes were then incubated with the corresponding peroxidase-coupled secondary antibody (anti-Rabbit, Abcam # 205718; anti-Mouse, Abcam # 205719, Cambridge, UK) diluted in 1% Tween-20 (*v*/*v*)/5% non-fat milk (*w*/*v*)/1×TBS, for 1 h at room temperature. Finally, after additional washes in 1% Tween-20 (*v*/*v*)/1×TBS, the membranes were incubated with peroxidase substrate (Thermo Fisher Scientific # 34094) and the signals visualized in an Omega Lum^TM^ G imaging system (Gel Company, San Francisco, CA, USA).

### 4.11. Preparation of Total RNA and RT-qPCR

Total RNA was extracted from adherent cells using TRIzol (Thermo Fisher Scientific, Waltham, MA, USA), according to the manufacturer’s instructions. RNA concentrations were determined automatically using a spectrophotometer (NanoDrop lite, Thermo Scientific, USA). One microgram of total RNA was used for complementary DNA (cDNA) synthesis. Reverse transcription requiring Oligo(dT)20 primer (Invitrogen, Waltham, MA, USA) was carried out using SuperScript III Reverse Transcriptase Kit (Invitrogen), following the manufacturer’s instructions. Quantitative Reverse Transcription PCR (qRT-PCR) assays were performed in technical triplicate, using Brilliant III Ultra-Fast SYBR^®^ Green QPCR master mix (Agilent, Santa Clara, CA, USA) and an Aria MX real-time qPCR instrument (Agilent, Santa Clara, CA, USA). The results were expressed as average (mean) fold-change compared to control samples, using the ΔΔCt method, in three biological experiments. Primers from the ribosomal gene L19 60S (RPL19) were used for internal calibration. The sequences of the primers used in this work are detailed in Table 1.

### 4.12. Harvesting of Conditioned Media

To collect conditioned media, 1 × 10^6^ cells were seeded on 10 cm diameter plates. The next day, cells began to be exposed to culture medium supplemented with one or more drugs, with replacement of the medium with the drug(s) after 48 h (this to ensure the maximum activity of the drug(s) during the entire exposure time). After 96 h of exposure, cells were washed three times in 1×PBS. After washing, 4 mL of serum-free medium was added to the plates. Then, 24 h later, conditioned media were collected in 2 mL Eppendorf tubes. Conditioned media were centrifuged for 5 min at 1000 rpm and 4 °C, and then stored in new tubes at −80 °C. After collecting ~25 mL per experimental condition, conditioned media were cold-thawed and concentrated 30-fold using Amicon Ultra 15 mL centrifugal filters (Merck Millipore Ltd., Tuliagreen, Carrigtwohill, Co Cork, Ireland). Aliquots of 50 μL of concentrated medium were stored at −80 °C until use.

### 4.13. Cytokine Analysis of Conditioned Media

A fraction of conditioned media, previously collected and concentrated (30-fold), was thawed and subjected to one analysis of flow cytometry using a Human Inflammatory Cytokine Cytometric Bead Array (CBA)-I kit (BD Biosciences, Franklin Lakes, NJ, USA), essentially as stated by the manufacturer’s instructions. Before cytometric analyses, total protein concentration was estimated for each aliquot of concentrated conditioned medium. Of note, protein quantification was necessary for the normalization of cytometry results. For functional assays (migration and invasion assays), each aliquot of conditioned medium was previously diluted in a 1:6 ratio with complete medium supplemented with 0.5% FBS.

### 4.14. Wound Healing Assays

Wound healing assays were performed in 24-well plates, as previously described [113]. Before seeding the cells, each well was coated with 10 µg/cm^2^ of Fibronectin for 1 h at 37 °C. After removing the unattached fibronectin, and washing twice with 1×PBS, 5 × 10^5^ cells were seeded on each well, in triplicate for each experimental condition. Then, 24 h later, when cells had grown to confluency, a wound was generated with a pipette tip, causing the disruption of the cell monolayer, a procedure that was followed by three washes with 1×PBS. Each cell monolayer, and its respective wound, was exposed to 1 mL of the respective conditioned medium supplemented with 0.5% FBS. Microscopic recordings and measurement of the wound closure were obtained for three fields per well at times 0, 12, 24, or 48 h.

### 4.15. Invasion Assays

Invasion assays were carried out using Transwell-type invasion chambers (Corning Inc., Corning, NY, USA) equipped with polycarbonate membranes (6.5 mm in diameter), each containing pores of 8.0 μM in diameter. Each membrane was treated with 10 μg/cm^2^ of Fibronectin for 1 h at 37 °C. After two washes with 1×PBS, 1 × 10^5^ MCF-7 or AGS cells, previously resuspended in 200 μL of serum-free medium, were added to the upper compartment of the Transwell chamber. In parallel, 500 μL of conditioned medium, diluted in 0.5% FBS-containing fresh medium (1:6 ratio), was added to the lower compartment of the Transwell chamber. After an incubation period of 6, 12, or 24 h (under 37 °C and 5% CO_2_ conditions), cells that had traversed the membrane and reached its lower surface were fixed in 70% ethanol for 10 min. After removing the ethanol, cells were stained with a 0.5% solution of Crystal Violet in methanol for 10 min. Five random images were captured with a 10x lens of an inverted bright light microscope (U-LS30-3, Olympus, Taichung, Taiwan), using the AmScope V3.7.7934 software. As a positive control, complete culture medium, supplemented with 10% FBS, was used as a chemoattractant at the lower compartment of the chambers. All experiments were performed in triplicate.

### 4.16. Statistical Analyses

Error bars represent the mean ± SEM of three independent experiments. **** *p* < 0.0001, *** *p* < 0.001, ** *p* < 0.01, * *p* < 0.05, ns = no significant difference compared to control (DMSO), and/or between each treatment. Analyses of two groups were performed using Student’s *t*-test. Analyses of more than two groups were based on two-way ANOVA with Tukey’s post-tests. Statistical analyses and plots were generated with the GraphPad Prism 9.0.1 program, using at least three replicates for each experiment.

## Figures and Tables

**Figure 1 ijms-24-09284-f001:**
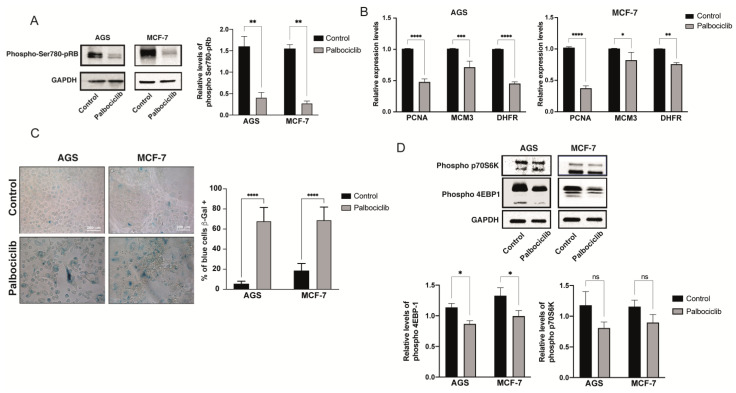
Palbociclib-driven senescence in AGS and MCF-7 cells: (**A**) Phosphorylation at Serine-780 of the retinoblastoma protein (pRB) following exposure of AGS and MCF-7 cells to Palbociclib. A representative Western blot and a densitometric analysis of phospho-Ser-780 of pRB are shown. (**B**) qRT-PCR-based expression levels of cell cycle-associated genes in AGS and MCF-7 cells after 96 h of exposure to Palbociclib. (**C**) AGS and MCF-7 cells were exposed to 1 μM or 0.5 μM of Palbociclib for 96 h, respectively. Cells were then subjected to senescence-associated β-galactosidase assay (SA-β-Gal) and then visualized with bright field microscopy at 40× magnification. Images are representative of at least three experiments, performed independently. (**D**) Phosphorylation levels of mTORC1 target proteins, 4EBP-1 and p70S6K, after exposure of AGS and MCF-7 cells to Palbociclib. Right, densitometry of the intensities of the bands corresponding to phospho-4EBP-1 and phospho-p70S6, and their comparison with the intensity of the corresponding GAPDH band. Images are representative of at least three independent experiments. Error bars represent the mean ± SEM of three independent experiments. **** *p* < 0.0001, *** *p* < 0.001, ** *p* < 0.01, * *p* < 0.05, ns = no significant difference compared to DMSO based on Student’s *t*-test.

**Figure 2 ijms-24-09284-f002:**
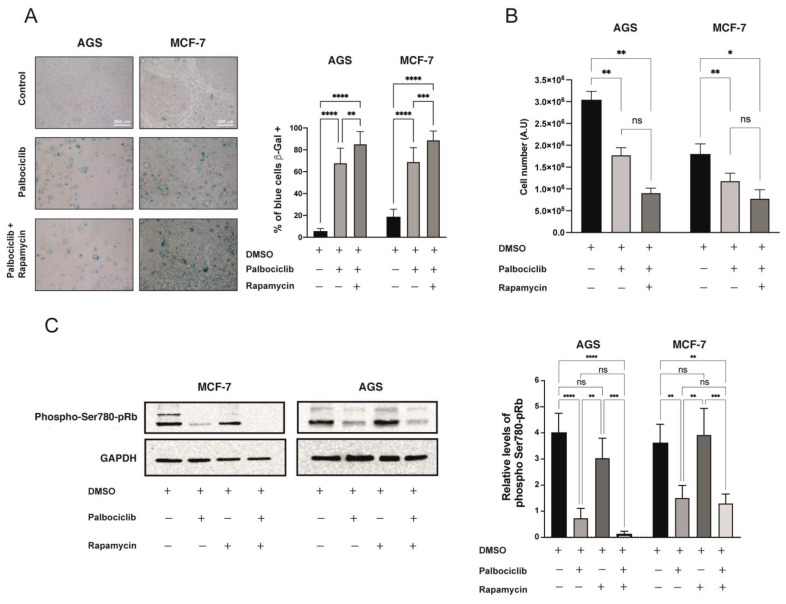
Senescence in cells treated with a combination of Palbociclib and Rapamycin: (**A**) AGS and MCF-7 cells were exposed to Palbociclib, or the combination of Palbociclib and Rapamycin, for 96 h. Cells were then subjected to Senescence-Associated β-Galactosidase assay (SA-β-Gal) and visualized with bright field microscopy at 40× magnification. Images are representative of at least three independent experiments. (**B**) AGS and MCF-7 cells were exposed to vehicle (DMSO), Palbociclib, or Palbociclib and Rapamycin, for 96 h. The number of viable cells was then quantified using the LUNA II counter. (**C**) The levels of phosphorylation in Serine-780 of the pRB (lower band) in cells previously exposed to the indicated experimental conditions were determined by Western blotting. Left: Densitometric analyses of the bands corresponding to phosphorylated Ser-780 of pRB, in comparison to the band of GAPDH. Error bars represent the mean ± SEM of three independent experiments. **** *p* < 0.0001, *** *p* < 0.001, ** *p* < 0.01, * *p* < 0.05, ns = no significant difference compared to DMSO based on two-way ANOVA with Tukey’s post-test.

**Figure 3 ijms-24-09284-f003:**
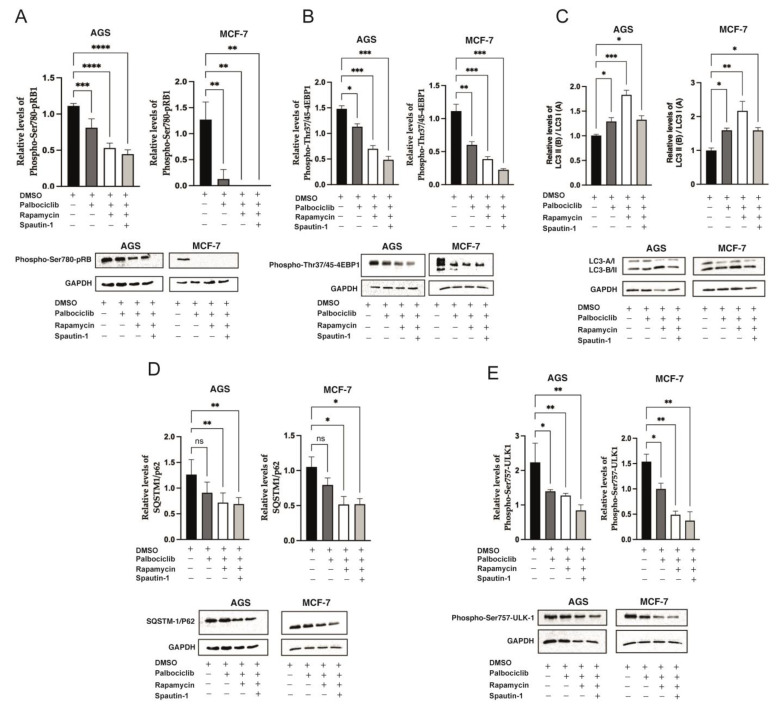
Autophagy levels in cells treated with a combination of Palbociclib, Rapamycin, and Spautin-1: AGS and MCF-7 cells were exposed to vehicle (DMSO), Palbociclib, the combination of Palbociclib and Rapamycin, or the combination of Palbociclib, Rapamycin, and Spautin-1, for 96 h. The levels of (**A**) phosphorylation at Threonine-37/46 of 4E-BP1 and phosphorylation at Serine-389 of p70S6K; (**B**) phosphorylation at Serine-780 of pRB; (**C**) LC3-I and LC3-II; (**D**) SQSTM-1/p62; and (**E**) phosphorylation at Serine-757 of the ULK-1, in cells previously exposed to the indicated experimental conditions, were determined by Western blotting. Left: Densitometric analyses of the respective bands compared to the band of GAPDH. Error bars represent the mean ± SEM of three independent experiments. **** *p* < 0.0001, *** *p* < 0.001, ** *p* < 0.01, * *p* < 0.05, ns = no significant difference compared to DMSO based on two-way ANOVA with Tukey’s post-test.

**Figure 4 ijms-24-09284-f004:**
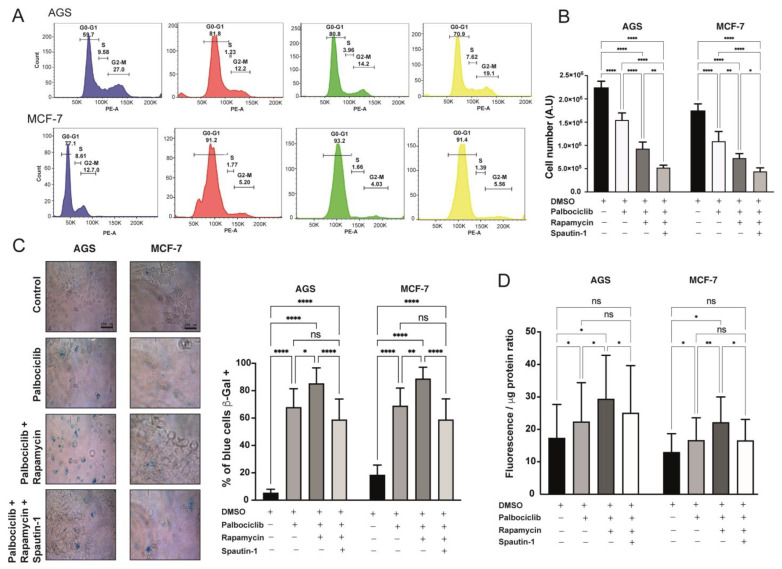
Cell cycle analyses of senescent cells subjected to Rapamycin, or combined Rapamycin and Spautin-1, treatment: (**A**) AGS and MCF-7 cells subjected to the indicated pharmacological treatments (in all cases, 96 h of treatment) were suspended in 1×PBS, fixed in ethanol, incubated with RNAse A, and stained with Propidium Iodide. Cells were then analyzed by flow cytometry in order to determine the relative proportion of cells in different phases of the cell cycle. (**B**) AGS and MCF-7 cells were exposed to the indicated drugs, or drug combinations, for 96 h. The number of viable cells was quantified with LUNA II counter. Error bars represent the mean ± SEM of three independent experiments. **** *p* < 0.0001, ** *p* < 0.01, * *p* < 0.05, based on two-way ANOVA with Tukey’s post-test. (**C**) AGS and MCF-7 cells were exposed to the indicated drug, or combination of drugs, for 96 h. Cells were then subjected to senescence-associated b-galactosidase assay (SA-β-Gal) and visualized with bright field microscopy at 40x magnification. Error bars represent the mean ± SEM of three independent experiments. **** *p* < 0.0001, ** *p* < 0.01, * *p* < 0.05, ns = no significant difference compared to DMSO based on two-way ANOVA. (**D**) Results of fluorometry-based β-galactosidase activity assays (ENZ-KIT129-0120, Enzo Life Sciences) following 96 h exposure of AGS and MCF-7 cells to the indicated drugs, or combination of drugs, are shown. Fluorescence values were corrected by the respective amounts of total protein. The results of at least three independent experiments were compiled. Error bars represent the mean ± SEM of three independent experiments, ** *p* < 0.0021, * *p* < 0.0332, ns *p* < 0.1234 (no significance) in comparison to DMSO treatment; based on two-way ANOVA with Geisser–Greenhouse correction and multiple comparisons with Tukey’s post-test.

**Figure 5 ijms-24-09284-f005:**
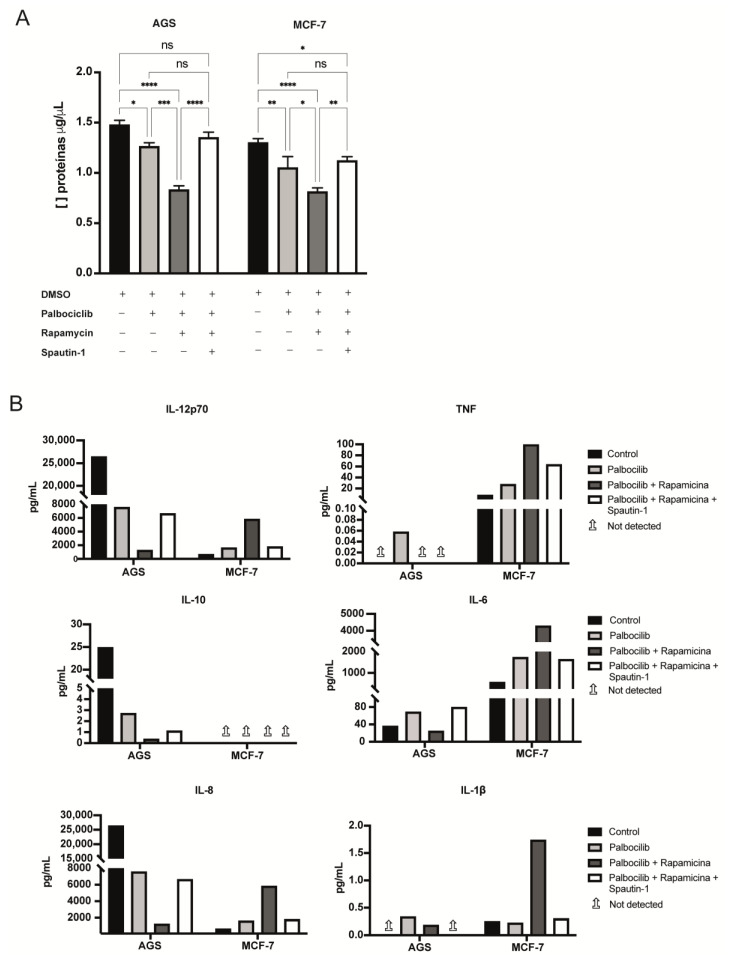
Detection of selected cytokines in conditioned media derived from senescent cells: (**A**) The concentration of total proteins in concentrated conditioned media derived from senescent cells exposed to the indicated drugs, or combination of drugs, was estimated using the Bradford assay. Error bars represent the mean ± SEM of two independent experiments. **** *p* < 0.0001, *** *p* < 0.001, ** *p* < 0.01, * *p* < 0.05, ns = no significant difference compared to DMSO based on two-way ANOVA. (**B**) Concentration of selected cytokines secreted by senescent cells subjected to different drug treatments. Arrows pointing up indicate that the molecule was not detected. The bar plots show a flow cytometry-based quantification (Human Inflammatory Cytokine Cytometric Bead Array) of the indicated cytokines detected in concentrated conditioned media derived from at least five cultures for each experimental condition.

**Figure 6 ijms-24-09284-f006:**
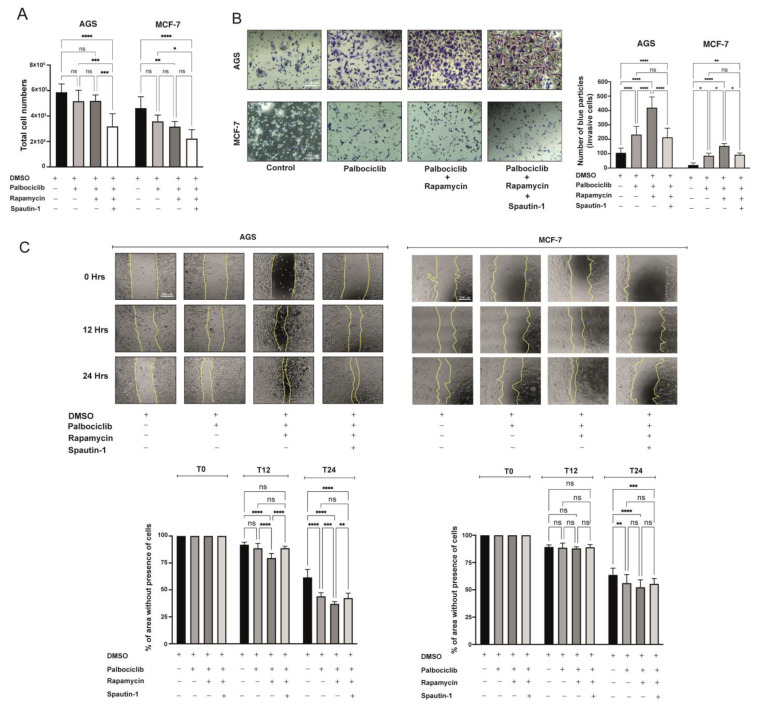
Paracrine effects of Palbociclib-driven senescent cells with or without inhibition of mTORC1, or with combined inhibition of mTORC1 and autophagy: (**A**) Non-senescent (actively proliferating) AGS and MCF-7 cells were exposed for 72 h to conditioned media derived from Palbociclib-driven senescent cells, Palbociclib-driven senescent cells with mTORC1 inhibition, or Palbociclib-driven senescent cells with combined mTORC1 and autophagy inhibition. All conditioned media were diluted in complete media supplemented with 0.5% FBS. The number of accumulated viable cells was determined by LUNA II counter. Error bars represent the mean ± SEM of three independent experiments. **** *p* < 0.0001, *** *p* < 0.001, ** *p* < 0.01, * *p* < 0.05, ns = no significant difference compared to DMSO, based on two-way ANOVA. (**B**) Non-senescent AGS and MCF-7 cells were exposed to conditioned media derived from Palbociclib-driven senescent cells, Palbociclib-driven senescent cells with mTORC1 inhibition, or Palbociclib-driven senescent cells with combined mTORC1 and autophagy inhibition. The ability of non-senescent AGS and MCF-7 cells to migrate through a fibronectin-coated porous polycarbonate membrane towards the lower compartment of a Transwell chamber (containing the conditioned medium) was assessed after 6–8 h of incubation. Images were captured at 10× magnification following the staining of the cells with a 0.2% Blue Violet solution. The images were further analyzed with the Image J Program. Results of three independent assays, in which at least eight microscopic fields per condition were analyzed, are shown. (**C**) Wound healing, migration, assays for non-senescent AGS and MCF-7 cells exposed to conditioned media derived from Palbociclib-driven senescent cells, Palbociclib-driven senescent cells with mTORC1 inhibition, or Palbociclib-driven senescent cells with combined mTORC1 and autophagy inhibition. All conditioned media were diluted in complete media supplemented with 0.5% FBS. Wound closure was assessed at time 0 (T0), and after 12 (T12), 24 (T24), and 48 (T48) hours of incubation in the respective conditioned medium. Images were captured at 10× magnification and further analyzed with the Image J Program. Results of two independent assays, performed in duplicate, were plotted. In each case, at least nine fields per condition were analyzed.

**Table 1 ijms-24-09284-t001:** Sequences of primers used.

RPL19	F: 5′-CAT CCG CAA GCC TGT GAC G-3′	R: 5′-TGT GAC CTT CTC TGG CAT TCG-3′
PCNA	F: 5′-GGA TAT TAG CTC CAG CGG TGT AAA-3′	R: 5′-TCT TCG GCC CTT AGT GTA ATG ATA-3′
MCM3	F: 5′-CCT TTC CCT CCA GCT CTG TCT AT-3′	R: 5′-GTG ATG GTC TGG TGA TCC TTG TAG-3′
DHFR	F: 5′-AAA CAA GGG GAA AGG GTT GGT TAG-3′	R: 5′-CCT CCC ATA TTG TCC CAG AGT AGT-3′
IL-8	F: 5′-AGG CAC AAA CTT TCA GAG ACA GCA G-3′	R: 5′-TGT TTA CAC ACA GTG AGA TGG TTC C-3′
BECLIN1	F: 5′-GGT GTC TCT CGC AGA TTC ATC-3′	R: 5′-TCA GTC TTC GGC TGA GGT TCT-3′
ULK-1	F: 5′-GGC AAG TTC GAG TTC TCC CG-3′	R: 5′-CGA CCT CCA AAT CGT GCT TCT-3′
PGC1a	F: AAC AGC AGC AGA GAC AAA TGC ACC-3′	R: 5′-TGC AGT TCC AGA GAG TTC CAC ACT-3′
DEC1	F: 5′-CCT TGA AGC ATG TGA AAG CA-3′	R: 5′-CAT GTC TGG AAA CCT GAG CA-3′
IL-6	F: 5′-GGC ACC TCA GAT TGT TGT TGT T-3′	R: 5′-GTG TCC TAA CGC TCA TAC TTT TAG T-3′
IL-1a	F: 5′-AGA TGC CTG AGA TAC CCA AAA CC-3′	R: 5′-CCA AGC ACA CCC AGT AGT CT-3′
IL-1b	F: 5′-ATG ATG GCT TAT TAC AGT GGC AA-3′	R: 5′-GTC GGA GAT TCG TAG CTG GA-3′

## Data Availability

Contact acayo@utalca.cl or nbrown@utalca.cl.

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
