# Peer review of "Palbociclib-Induced Cellular Senescence Is Modulated by the mTOR Complex 1 and Autophagy"

_ijms, 2023, doi:10.3390/ijms24119284_

Round 1
Reviewer 1 Report
In this study, the authors demonstrated that mTORC1 inhibition exacerbates Palbociclib-induced cellular senescence, which is mediated by autophage induction. Overall, this study was well-designed and experiments were performed rigorously. Conclusions are fully supported by the existing results. I only have some minor concerns below.
1, They claimed a further decrease in the levels of p-pRB in the mTORC1 inhibition group, however, the blots and quantifications showed no difference in Figure 2C. The authors should clarify this.
2, The graphs in Figure 5B showing cytokine concentrations have no error bar and no statistic analysis. The authors need to confirm that how many replicates do they have for this experiment and how do they make the conclusion without statistic analysis.
This manuscript is well-written. Only minor editing of language is required.
Author Response
Reviewer #1.
In this study, the authors demonstrated that mTORC1 inhibition exacerbates Palbociclib-induced cellular senescence, which is mediated by autophagy induction. Overall, this study was well-designed, and experiments were performed rigorously. Conclusions are fully supported by the existing results. I only have some minor concerns below.
- They claimed a further decrease in the levels of p-pRB in the mTORC1 inhibition group, however, the blots and quantifications showed no difference in Figure 2C. The authors should clarify this.
- Response: The reviewer is correct. Even though we pointed out the phenomenon qualitatively, we failed to find significant differences when the intensity of the phospho-bands was quantified and compared to the group of Palbociclib-treated cells. In an attempt to clarify this concern, we have incorporated the following paragraph into the manuscript: “Interestingly, this increase in senescent cells after treatment with Palbociclib and Rapamycin was accompanied by a further reduction in the phosphorylated levels of pRB at Serine-780, an effect that was particularly marked in MCF-7 cells (Figure 2C). Nevertheless, this trend did not reach significance when compared with the levels phospho-pRB in cells treated with Palbociclib alone (Figure 2C). Therefore, mTORC1 inhibition seems to exacerbate the pro-senescence effect of Palbociclib”. Having said that, we do find the phenomenon mechanistically relevant, as it may explain, at least in part, the exacerbation in the senescent phenotype in senescent cells exposed to Rapamycin.
- The graphs in Figure 5B showing cytokine concentrations have no error bar and no statistical analysis. The authors need to confirm that how many replicates do they have for this experiment and how do they make the conclusion without statistical analysis.
- Response: The experiments shown in Figure 5B are complex and labor-consuming. As explained in the section of Materials and Methods, they involved the collection of large volumes of conditioned media, derived from multiple culture plates, for each experimental condition. For each condition, media were then concentrated 30-fold, obtaining a volume enough to be utilized in one bead-based assay. Therefore, although we were evaluating multiple conditioned media for each condition since they were combined, we were unable to do a proper statistical analysis. Nevertheless, we believe that these experiments are still valuable when interpreted in the context of what we had learned from the findings shown in Figures 1 to Figure 4: they showed that inhibition of mTORC1 tended to exacerbate senescence-associated secretory responses (further reducing secretion in senescent AGS cells or further increasing secretion in senescent MCF-7 cells) and, more interestingly, these responses were reversed by Spautin-1, highlighting the relevance of the mTORC1-autophagy axis as a regulator of SASP components in different cell lineages.
Reviewer 2 Report
This study makes a significant contribution to the understanding of the complex relationship between cellular senescence, mTORC1 activity, and autophagy, especially in the context of therapy-induced senescence. The researchers have examined these aspects using Palbociclib-induced senescent AGS and MCF-7 cells, providing new insights into the cellular mechanisms of senescence and the senescence-associated secretory phenotype (SASP). The study’s key strengths lie in its experimental design and the scope of the investigation. The authors have adopted a multi-faceted approach to investigate the effects of mTORC1 and autophagy inhibition on Palbociclib-driven senescence, providing a more comprehensive view of these interconnected processes. The observation that further mTORC1 inhibition exacerbated the senescent phenotype, which could be reversed by autophagy inhibition, is particularly intriguing and highlights the intricate balance of these cellular mechanisms. The finding that the SASP profile changes upon inhibiting mTORC1 or combined inhibition of mTORC1 and autophagy, influencing cell proliferation, invasion, and migration of non-senescent tumorigenic cells, has important implications for cancer biology and therapy. However, these effects need to be validated in more complex biological systems and in vivo models, as the study is primarily based on in vitro observations. While the study is overall well-conducted, it would benefit from a more detailed discussion on the potential applications of these findings in clinical settings. For example, how might these results influence the development of novel cancer treatment strategies or the optimization of existing therapies? Moreover, a clear explanation of the study's limitations, such as potential effects of cell line-specific characteristics on the findings, would strengthen the manuscript. In conclusion, this study provides a valuable exploration of the interplay between mTORC1 signaling, autophagy, and the SASP in the context of CDK4/6 inhibitor-induced senescence. Further research is needed to fully understand these complex mechanisms and translate these findings into clinical practice.
1. There are a few instances where the sentence structure could be improved. For instance, in the sentence "Notably, mTORC1 activity is also necessary for the implementation of cellular senescence, at least in some models." it would be better to say "Notably, in some models, mTORC1 activity is also necessary for the implementation of cellular senescence." There seem to be a few typos, such as "un upregulation of autophagy".
2.Make sure to define all abbreviations at their first occurrence. While most of them have been defined, there are a few that haven't been explained, such as EMT (Epithelial-Mesenchymal Transition), OIS (oncogene-induced senescence), and TASCC (TOR Autophagy Spatial Coupling Compartment).
3.Drug-induced autophagy is a process where drugs are used to activate the body's cellular recycling system to eliminate damaged or harmful components. It should be update in the introduction section, such as PMID: 33392197, 33838688.
4.While some discrepancies and potential explanations are mentioned, it would be beneficial to formally acknowledge the limitations of your study. This could involve discussing the limitations of your cell models, or potential issues with the methods used.
5.Your discussion could be enhanced by providing clear suggestions for future research based on your results. For example, you mention the need to confirm a hypothesis regarding CDK4/6 inhibition and reduced D-type cyclin synthesis, but further explicit suggestions would be beneficial.
6.The table one should be moved to the supplementary data.
7.Two bands are observed in the phospho-Rb (Ser780) analysis; determining which one accurately represents the phosphorylated Rb protein is essential.
8.The microscopy photos lack a scale bar. To improve the accuracy and usefulness of these photos, it's important to include a scale bar in the microscopy images.
9. The font size in the figures is too small to read, which can hinder the audience's ability to comprehend the information being presented. It's essential to ensure that figures are presented in a clear and legible manner to effectively communicate research findings. To address this issue, the font size should be increased, or the figures can be re-designed to better display the data. By improving the visibility of the figures, the audience can more easily interpret the results and gain a better understanding of the research.
Minor editing of English language required
Author Response
- However, these effects need to be validated in more complex biological systems and in vivo models, as the study is primarily based on in vitro observations. While the study is overall well-conducted, it would benefit from a more detailed discussion on the potential applications of these findings in clinical settings. For example, how might these results influence the development of novel cancer treatment strategies or the optimization of existing therapies? Moreover, a clear explanation of the study's limitations, such as potential effects of cell line-specific characteristics on the findings, would strengthen the manuscript.
- Response: We thank the reviewer for the recognition of the merits of our work. We are well aware of the limitations of any in vitro study utilizing cell lines and, therefore, we agree that further in vivo work must be done in order to demonstrate that Palbociclib-induced senescence, and its associated SASP profiles, can be modulated through inhibition of mTORC1 and/or autophagy in order to maximize the therapeutic benefits of Palbociclib and other pro-senescence cancer therapies. Since Palbociclib is already utilized in patients with breast cancer, further in vivo studies should focus on responses of mammary tumors arising in mouse models of breast cancer to Palbociclib, with or without inhibition of mTORC1 and/or autophagy. To be more translatable to the clinic, experiments of transplantation of organoids or human tumor tissue derived from patients with breast cancer should be used as a starting point. These concerns have been addressed at the beginning of the Discussion section.
- There are a few instances where the sentence structure could be improved. For instance, in the sentence "Notably, mTORC1 activity is also necessary for the implementation of cellular senescence, at least in some models." it would be better to say "Notably, in some models, mTORC1 activity is also necessary for the implementation of cellular senescence." There seem to be a few typos, such as "un upregulation of autophagy".
- Response: We apologize for the unintended grammatical errors. As suggested by the reviewer, we have introduced the recommended changes.
- Make sure to define all abbreviations at their first occurrence. While most of them have been defined, there are a few that haven't been explained, such as EMT (Epithelial-Mesenchymal Transition), OIS (oncogene-induced senescence), and TASCC (TOR Autophagy Spatial Coupling Compartment).
- Response: We have done a thorough re-reading of the manuscript in order to detect these and other errors of format. As suggested by the reviewer, we have defined all abbreviations in the manuscript the first time they are mentioned.
- Drug-induced autophagy is a process where drugs are used to activate the body's cellular recycling system to eliminate damaged or harmful components. It should be update in the introduction section, such as PMID: 33392197, 33838688.
- Response: As suggested by the reviewer, the sentence was added.
- While some discrepancies and potential explanations are mentioned, it would be beneficial to formally acknowledge the limitations of your study. This could involve discussing the limitations of your cell models or potential issues with the methods used.
- Response: We are well aware of the limitations of any in vitro study utilizing cell lines and, therefore, we agree that further in vivo work must be done in order to demonstrate that Palbociclib-induced senescence, and its associated SASP profiles, can be modulated through inhibition of mTORC1 and/or autophagy in order to maximize the therapeutic benefits of Palbociclib and other pro-senescence cancer therapies. Since Palbociclib is already utilized in patients with breast cancer, further in vivo studies should focus on responses of mammary tumors arising in mouse models of breast cancer to Palbociclib, with or without inhibition of mTORC1 and/or autophagy. To be more translatable to the clinic, experiments of transplantation of organoids or human tumor tissue derived from patients with breast cancer should be used as a starting point. In order to convey this message, we inserted the following paragraph at the beginning of the Discussion:
“In this work, we study the functional consequences of inhibiting mTORC1, or mTORC1 and autophagy, on senescent cells driven by Palbociclib, an inhibitor of CDK4/6. To this end, we took advantage of MCF-7 (breast carcinoma) and AGS (gastric adenocarcinoma) cells, both of which retain a functional pRB pathway and are responsive to CDK4/6 inhibition. While the initial development or assessment of anti-cancer drugs require the use of cell lines, animal models, especially patient-derived xenograft mouse models, are an essential bridge between preclinical studies and clinical trials in cancer treatment (89). In this context, we are aware of the limitations of using an in vitro model and we recognize the need to confirm our findings using in vivo models. Nevertheless, our results clearly suggest the therapeutic relevance of modulating drug-induced senescence”.
Your discussion could be enhanced by providing clear suggestions for future research based on your results. For example, you mention the need to confirm a hypothesis regarding CDK4/6 inhibition and reduced D-type cyclin synthesis, but further explicit suggestions would be beneficial.
Response: In order to accommodate the reviewer’s suggestion, we have introduced the following paragraph towards the end of the Discussion section: “Overall, our findings demonstrate the relevance of redirecting the secretory profiles of senescent cells through pharmacological intervention. More specifically, the fact that some functional features of Palbociclib-driven senescent cells, including the SASP, can be modulated by drugs that block mTORC1 or autophagy, offers new ways to enhance the beneficial, anti-proliferative, effects of inhibiting CDK4/6 in tumors with molecular evidence of high CDK4/6 activity. At the same time, a note of caution must be added before these combinations of drugs can be introduced into clinical practice, as the secretome of Palbociclib-driven senescent cells could enhance the migratory and invasive capabilities of cancer cells that might have escaped senescence”.
- The table one should be moved to the supplementary data.
- Response: As suggested by the reviewer, we have moved Table I to the Supplementary Data Section.
- Two bands are observed in the phospho-Rb (Ser780) analysis; determining which one accurately represents the phosphorylated Rb protein is essential.
- Response: In an attempt to answer this concern, we have added, in the legend of Figure 2, a sentence in which we make clear that the band we are referring to as phosphorylation of Ser-780 of pRB is the lower band.
- The microscopy photos lack a scale bar. To improve the accuracy and usefulness of these photos, it's important to include a scale bar in the microscopy images.
- Response: As suggested by the reviewer, we have inserted scale bars into the microscopic images.
- The font size in the figures is too small to read, which can hinder the audience's ability to comprehend the information being presented. It's essential to ensure that figures are presented in a clear and legible manner to effectively communicate research findings. To address this issue, the font size should be increased, or the figures can be re-designed to better display the data. By improving the visibility of the figures, the audience can more easily interpret the results and gain a better understanding of the research.
- Response: We apologize for the poor quality of the images displayed. We have made changes in font size for all the figures.
Round 2
Reviewer 2 Report
The authors have address all of my comments and the manuscript improved significantly.